# Characterization of Loss-of-Imprinting in Breast Cancer at the Cellular Level by Integrating Single-Cell Full-Length Transcriptome with Bulk RNA-Seq Data

**DOI:** 10.3390/biom14121598

**Published:** 2024-12-14

**Authors:** Muhammad Talal Amin, Louis Coussement, Tim De Meyer

**Affiliations:** 1Department of Data Analysis and Mathematical Modelling, Faculty of Bioscience Engineering, Ghent University, 9000 Ghent, Belgium; muhammadtalal.amin@ugent.be (M.T.A.); louis.coussement@ugent.be (L.C.); 2Cancer Research Institute Ghent (CRIG), 9000 Ghent, Belgium; 3Department of Bioscience and Technology, Khwaja Fareed University of Engineering and Information Technology, Rahim Yar Khan 64200, Pakistan; 4Bioinformatics Institute Ghent N2N, Ghent University, 9000 Ghent, Belgium

**Keywords:** Genomic imprinting, single-cell RNA-seq, bulk RNA-seq, loss-of-imprinting (LOI), breast cancer

## Abstract

Genomic imprinting, the parent-of-origin-specific gene expression, plays a pivotal role in growth regulation and is often dysregulated in cancer. However, screening for imprinting is complicated by its cell-type specificity, which bulk RNA-seq cannot capture. On the other hand, large-scale single-cell RNA-seq (scRNA-seq) often lacks transcript-level detail and is cost-prohibitive. Here, we address this gap by integrating bulk RNA-seq with full-length transcript scRNA-seq to investigate imprinting dynamics in breast cancer. By analyzing scRNA-seq data from 486 cancer cells across subtypes, we identified multiple SNPs in imprinted genes, including *HM13*, *MEST* (*PEG1*), *SNHG14* and *PEG10*, showing consistent biallelic expression. Bulk RNA-seq, however, revealed that this biallelic expression arises from transcript-specific imprinting, rather than loss-of-imprinting (LOI). The imprinted SNPs identified in bulk RNA-seq predominantly demonstrate proper monoallelic expression in scRNA-seq. As a clear exception, an HER2+ breast cancer sample exhibited distinct LOI of *MEST*. Previous bulk RNA-seq-based observations about *MEST* LOI in breast cancer could not exclude a non-cancer cell impact, but our results validate that *MEST* LOI is cancer-specific. This study demonstrates the complementary utility of bulk and scRNA-seq in imprinting studies, confirming *MEST* LOI as a genuine event in breast cancer.

## 1. Introduction

Genomic imprinting is an epigenetic phenomenon that causes genes to express a single allele in a parent-of-origin-specific manner (Figure 1). Imprinting is not fully static, as it may vary across tissues and developmental stages [1]. Moreover, it is sometimes transcript-specific, meaning that different transcripts of the same gene can exhibit distinct parent-of-origin expression patterns, with some being paternally expressed and others maternally expressed [2] (Figure 1). In line with its role in growth control, where particularly paternally expressed imprinted genes promote growth, its disruption has been widely implicated in cancer development [3]. Specifically, loss of imprinting (LOI), where the normally silenced allele is expressed again, is a common phenomenon in various cancers. For example, breast cancer frequently features LOI in genes such as *IGF2*, *HM13*, *MEST* and *MEG3* [4]. Moreover, imprinting deregulation has clinical implications, leading to increased cancer risk or aggressiveness [5]. However, most studies have focused on DNA methylation, typically using Infinium HumanMethylation BeadChips, at imprinting control regions, yet do not provide direct evidence for LOI. Therefore, more profound research is required to elucidate the role of LOI in cancer.

However, the first major bottleneck is the overall cost of LOI studies. LOI is typically detected by studying gene expression in bulk RNA-seq data and comparing the expression levels of the two alleles of an imprinted gene. This strategy depends on the existence of single-nucleotide polymorphisms (SNPs) or indels that can differentiate between alleles in heterozygous individuals. Hence, such studies require large-scale sequencing data to ensure sufficiently large numbers of heterozygous individuals for the locus under study, particularly when the frequency of LOI is relatively low. Although, originally, DNA data were also required to identify heterozygous individuals per locus, a more recent strategy enables the detection of (loss-of) imprinting, starting solely from RNA-seq [4].

But even then, bulk RNA-seq still has its limitations, as it does not capture the specific cell type or state that exhibits (loss-of) imprinting, leading to a second important bottleneck. Indeed, even though LOI entails the expression of the normally silenced allele, it does not always go hand in hand with overall expression upregulation. This is because some genes, such as *MEST* LOI in breast cancer, appear to coincide with expression downregulation [4], casting doubt on the assumption that LOI invariably leads to increased gene expression. Several confounders have been proposed, such as residual biallelic expression in admixing non-cancer cells, which becomes clear upon the downregulation of the gene in cancer, but also the impact of global chromatin changes in cancer cells [6,7].

The allele-specific expression analysis of single-nucleus or single-cell RNA-seq (scRNA-seq) may overcome these bulk RNA-seq limitations and reveal the heterogeneity and dynamics of LOI at a single cell resolution [8,9]. For example, Federico et al. validated the imprinting status of known imprinted genes expressed in fibroblasts and discovered nine new imprinted genes, demonstrating the advantages of single-cell RNA-seq over bulk RNA-seq in identifying imprinted genes [10]. However, most scRNA-seq experiments only target UTRs, and thus do not provide the transcript-level resolution required for imprinting analysis [11], explaining a general lack of scRNA-seq studies focusing on LOI in cancer cells. An additional challenge of the scRNA-seq strategy when studying imprinting is that it is currently too expensive for the large-scale discovery of (loss-of) imprinting, and ideally also requires genotyping data [12].

Here, we present a novel strategy for studying LOI by integrating bulk-RNA-seq and full-length single-cell RNA analyses. This approach enables the identification of tumor samples that feature LOI in specific genes and demonstrate the genuine LOI of *MEST* in breast cancer.

## 2. Materials and Methods

### 2.1. Data Collection

Full-length single-cell RNA sequencing data (SMART-seq) from 549 primary breast cancer cells in 11 patients, along with bulk RNA data from the same patients, were gathered from research conducted by Chung et al. [13]. Cell-level and bulk data from this study are publicly accessible at the ENA (European Nucleotide Archive) under the accession number PRJNA305054. The dataset includes various breast cancer subtypes: luminal A (N = 2), luminal B (N = 1), HER2+ (N = 3) and triple-negative breast cancer (TNBC) (N = 5). Additionally, regional metastatic lymph node data from one luminal B and one triple-negative breast cancer patient were included in the study. For one sample (BC09), the scRNA-seq experiment had been run in duplicate, and these data were merged for further preprocessing. Details of the data are provided in the Appendix A.

### 2.2. scRNA-Seq Data Preprocessing

scRNA-seq data were mapped to the reference genome GRch38 using the STAR aligner (version 2.7.10a) [14], relying on the GENCODE database (release 44) for gene annotation. As SMART-seq does not rely on unique molecular identifiers (UMI’s), Picard’s MarkDuplicates tool (version 2.27.4) was used to avoid biases or artefacts introduced by PCR duplicates [15]. Upon deduplication, we performed base quality score recalibration using GATK’s BaseRecalibrator (version 4.3.0.0) and ApplyBQSR (version 4.3.0.0) tools. This process adjusted the quality scores of the sequencing reads to correct for systematic biases, using known variant sites from a reference database (dbsnp138) to guide the recalibration. The resulting high-quality BAM files (filtered on a recalibrated phred score of at least 25) were then used for precise allele counting and as the basis for cell type clustering.

### 2.3. Allele Count Quantification and Lesser Allele Fraction (LAF)

In our analysis, samtools mpileup (version 1.15.1) was used to generate mpileup files from the BAM files [16]. These mpileup files provide a detailed view of the base calls and their quality at each genomic position. We then used a custom Perl script to process these mpileup files, extracting allele-specific counts based on the dbSNP locations (dbsnp138). For all cancer and non-cancer cells, the counts per allele are available in the Appendix A for SNPs with at least one count.

Following the extraction, we annotated the allele-specific counts with rsIDs from the variants database, dbSNP, and with gene identifiers using ANNOVAR (version 2018Apr16) with GRch38 [17]. To enhance the interpretability of our variant data, gene and rsID information was incorporated into the base call files. Finally, we parsed the annotated base call files to generate annotated high-quality files of the counts per allele for known SNPs.

Relying on these counts, we calculated the lesser allele fraction (LAF) per SNP and individual sample, defined as the ratio of the smaller allele count (either reference or alternative) to the total allele count (sum of reference and alternative counts), thereby providing a measure of allelic imbalance.

### 2.4. Clustering and Cell Type Identification

FeatureCount (version2.0.1) with the hg38 GTF was used on the cell-specific BAM files for expression quantification per gene and cell. Seurat (version 5.0.3) was used for quality control and further data processing. Only those cells that exhibited a total number of genes between 200 and 20,000 and > 200 RNA molecules were kept. The feature counts for each cell were divided by the total counts for that cell and multiplied by a scale factor of 10,000 before performing a log transformation using Seurat’s NormalizeData function [18]. Only genes expressed in at least 10% of cells were retained, and the top 2000 variable features were used for clustering. All remaining cells were subsequently merged into a single dataset and batch normalization and scaling was performed using the harmony package (version 1.2.0) to correct for technical variability due to batch effects, followed by PCA (Principal component analysis) and dimensionality reduction using Seurat’s UMAP functionality.

Marker genes were collected from the literature and the reference article for the data at hand [13]. To refine our gene list for cell type identification, we combined the top variable features with our predefined markers, ensuring that the final list contained genes with sufficient discriminatory power to distinguish between different cell types. The scSorter algorithm (version 0.02) was then applied to the subsetted expression data [19], which allowed us to predict cell types and subtypes based on the expression of marker genes.

Cell type annotation information was then merged with the allelic counts data, yielding the final cell dataset. For cluster fine-tuning, we ran UMAP with adjusted parameters, specifically an increased number of neighbors = 20, a higher minimum distance = 0.5, and using the cosine as the distance metric, to optimize the granularity of our cell type distinction.

### 2.5. LOI Detection in scRNA-Seq Data

For LOI detection, we used a one-sided binomial test that evaluated whether the least expressed allele was more frequently detected than expected based on the anticipated sequencing error rate. Here, we set the expected sequencing error rate at 0.5%, which is a conservative estimate given the median error rate <0.2% for the HiSeq2500 system used here [20]. We only statistically evaluated sufficiently covered SNPs, i.e., with a total count ≥4 in the cell under study (Appendix A). *p*-values underwent False Discovery Rate (FDR) adjustment using the Benjamini–Hochberg method; this was performed across all cells and sufficiently covered SNPs identified as imprinted in the reference paper. Moreover, we additionally filtered significant results so that only cells with at least two or more counts for both alleles in a normally imprinted SNP were considered to exhibit biallelic expression (LOI).

### 2.6. Bulk RNA Data

A similar preprocessing strategy was used to generate allele counts per SNP from the bulk RNA data generated for the same 11 patient samples, relying on alignment, data summary and base quality recalibration and filtering methods and settings, as outlined for scRNA-seq. Alignment details are provided in Appendix A. The same strategy was used for LOI detection, as described for the single-cell data, but per patient rather than per cell.

### 2.7. Custom Scripts

Alongside to the software packages mentioned higher, custom scripts in R (version 4.3.0), Perl (version 5.32.1) and Python (version 3.10.6) were used for parsing and similar low-level data processing. All scripts can be accessed at the following GitHub repository: https://github.com/talalamin/LOI-SCALE.

## 3. Results

The strategy we used to identify tumor-specific LOI starting from scRNA-seq data is outlined in Figure 2. Initially, the full-length single-cell dataset generated by Chung et al. (SMART-seq for 549 primary breast cancer cells in 11 patients) is processed to generate two distinct datasets per cell: one for identifying biallelic expression by counting alleles per SNP, and another for gene-level expression for cell annotation [13]. For putatively imprinted genes and SNPs identified from the bulk sample sequencing strategies used in the breast (here obtained from [4]), LOI in cancer is subsequently identified as the biallelic expression of imprinted SNPs in single cancer cells (see Methods), which may corroborate the identification of LOI using bulk RNA-seq-driven approaches. On the other hand, biallelic expression in non-cancer cells disproves the role of LOI for this gene in the cancer type under study. Additionally, for putatively imprinted genes, SNPs biallelically expressed in both single-cell cancer and bulk normal RNA-seq data may also indicate the presence of transcript-specific imprinting rather than LOI in cancer. Indeed, in the case of transcript-specific imprinting, some transcripts are maternally and others paternally expressed, but these may still share exons with SNPs that hence exhibit biallelic expression (cf. Figure 1C). Additionally, though not an essential part of this strategy, we verified LOI in bulk RNA-seq data also provided by Chung et al. [13] for the patient samples at hand. In summary, this strategy ensures a more robust identification of LOI by integrating single-cell and bulk RNA-seq data.

### 3.1. Data Pre-Processing and Cell Type Annotation

After read alignment (median alignment rate of 83%, Appendix A) and base quality score recalibration, the high-fidelity allele counts per cell for known variant sites were extracted from the BAM files. Concurrently, preprocessing and quality filtering of the 549 single cell expression profiles were performed (cell gene/transcript distributions per patient are provided in Figure 3). As technical batch effects were present (Appendix A), the Harmony algorithm was used for correction, resulting in biologically meaningful clustering (Appendix A).

UMAP visualization (of 486 remaining cells) with marker gene information was used for cell type annotation. The clustering revealed distinct cell populations, including B cells, CD8+ NKT-like cells, dendritic cells, endothelial cells, hematopoietic stem/progenitor cells (HSC/MPP), ISG-expressing immune cells, macrophages, mast cells, monocytes, T cells and tumor subtypes (luminal A, luminal B, HER2+ and TNBC). Single cells with ambiguous marker gene inferred annotation were labelled as unknown cells (Table 1, Figure 4).

### 3.2. Single-Cell Level Analysis of (Loss-Of) Imprinting

After merging cellular annotation with SNP variant information, we screened the single-cell RNA data for the bi- and monoallelic expression of SNPs for a total of 29 (putatively) imprinted genes reported in bulk breast RNA studies [4] (Appendix A). The raw count data used for screening are provided as a Appendix A. If imprinting is present, we would expect the expression of only a single allele for all SNPs of that gene, with biallelic expression indicative of putative LOI. Note that the opposite does not automatically hold true, i.e., SNPs for which only a single allele is detected may still feature (loss-of) imprinting when the patient is homozygous for that SNP. As homozygosity inherently masks parent-of-origin-specific expression patterns, LOI and imprinting can only be detected at heterozygous SNPs.

A large number of genes (Appendix A), including well-known imprinted ones such as *MEST*, *H19*, *HM13*, *MEG3*, *PEG10*, *PEG3*, *SNHG14*, *PWAR6* and *SNRPN*, featured SNPs with clear biallelic expression in multiple cancer cells, which is compatible with the basic definition of LOI. However, we hypothesized that this could be attributed to transcript-specific imprinting, where SNPs located in exons shared between paternally and maternally expressed transcripts may feature biallelic expression (cf. Figure 1C). Though our SMART-seq scRNA-seq data provide an overview of the SNPs present across the full length of the transcript, the Illumina short reads do not readily enable transcript-level analyses. We therefore matched the single-cell dataset with the list of putatively imprinted SNPs in healthy breast tissue from our previous bulk RNA-seq study (Illumina, reads generated along entire length of cDNA sequences) [4]. This revealed that the large majority of biallelically expressed SNPs in the scRNA-seq data are indeed derived from non-imprinted SNPs/transcripts. Hence, to discriminate LOI from normal transcript-specific biallelic expression in single-cell data, a comprehensive overview of normal imprinting at a SNP level resolution is required, which was here obtained from the bulk RNA-seq results. Upon selecting known imprinted SNPs, only six gene–sample combinations (for 6 SNPs) that featured putative LOI in the single-cell RNA-seq data remained (Appendix A); these were further subjected to in-depth allele-specific expression analysis.

In a HER2+ tumor sample (patient BC05), the single *MEST* SNP (rs10863) reported as imprinted in healthy bulk breast tissue exhibited significant (FDR < 0.05) LOI in no less than 9 out of the 12 (75%) tumor cells meeting the filtering criteria for this patient (Figure 5A, Table 2, Appendix A). LOI was further verified by the visual inspection of reads aligned to the SNP (Appendix A). Moreover, the LOI of this SNP was clearly supported by bulk RNA-seq for the same tumor sample (Table 2, Appendix A). Of interest, the same SNP was reported to feature significant LOI in HER2+ breast cancer in a previous bulk RNA-seq study (Table 2, Appendix A) [4]. In addition, three non-tumor cells were identified in this sample (BC05), including one classified as an immune T cell and two unknown cells. We observed no expression of *MEST* in the T-cell and one unknown cell, and only a single count for one allele in the other unknown cell (see Appendix A, sheet sc BC05). Moreover, there was no major downregulation of *MEST* in this tumor sample (BC05, Figure 5B), discarding downregulation as a cause of (higher) biallelic expression (see Section 4). We also examined the available copy number alteration data for this patient [12], finding no alterations for the *MEST* region.

Of interest, the same, normally imprinted, *MEST* SNP (rs10863) also showed biallelic expression in the bulk RNA data of a luminal A tumor (BC01, Appendix A). This was not directly observed in the available single-cell data from the same patient: of the 26 cells available, only 4 cells featured sufficient expression for this SNP. This indicates that a higher number of single cells and/or deeper sequencing may be required to detect clear LOI. Additionally, *PTX3* (rs73158510) is a putatively imprinted locus that showed biallelic expression in three TNBC cancer cells (patient BC11), corroborated by the bulk RNA (LOI) results for the closely related basal-like subtype [4]. This biallelic expression was also evident in the bulk RNA sample of the same individual (Table 2, Appendix A).

An imprinted *ZDBF2* SNP (rs4673350) showed LOI in a single cancer cell from a HER2+ patient (BC05) but was not detected in the bulk data for the same individual (Table 2). Similarly, two cancer cells from TNBC patient BC06 showed clear biallelic expression (lowest expressed allele detected 4 and 19 times, highest 19 and 53 times) for *PEG10* (rs13073), which was not detected in the accompanying bulk sample. Though this can be attributed to the limitations of bulk RNA-seq for the detection of LOI when only occurring in a small fraction of tumor cells, this makes clinical relevance in this sample unlikely (Table 2). Nevertheless, in both cases, independent bulk RNA-seq studies supported the imprinting of the SNP in breast tissue [4], but did not clearly evidence that LOI plays a major role in the breast cancer subtype under study (using basal-like cancer as a proxy for the TNBC considered here). Therefore, to enable the interpretation of LOI events in single-cell data, it is necessary to include previously large-scale bulk-derived (loss-of) imprinting information to evaluate their relevance.

However, it is clear that integrating single-cell data also provides major benefits, for example, by demonstrating that biallelic expression/LOI is clearly present in tumor cells (as observed for *MEST*) rather than in the normal cells infiltrating/present in the tumor mass. This is particularly relevant for genes featuring random monoallelic expression (RME): such genes often show a major bias towards a single random (rather than parent-of-origin specific) allele, which leads to an allele-specific expression profile very similar to an imprinted gene with some minor LOI. For example, several SNPs (rs241402 and rs6912492) at *LOC100294145* (candidate imprinted gene according to [4]) exhibited biallelic expression in multiple single cells of a luminal B metastatic tumor sample (BC03 LN), and biallelic expression was also detected in the bulk RNA sample for the same patient (Table 2, Appendix A). However, at the single-cell level, biallelic expression was only observed in immune B-cells (Figure 6). As this gene is located in the human leukocyte antigen (HLA) cluster with multiple RME genes, the results presented here clearly demonstrate that RME is a far more likely explanation for *LOC100294145* than (loss-of) imprinting. Similarly, SNP rs241408 of the same gene (*LOC100294145*) exhibited biallelic expression in a TNBC bulk RNA patient (BC09) (Appendix A), even though it was not visible in single cells. This emphasizes the importance of integrating bulk RNA with single-cell RNA data to better distinguish cancer-specific LOI from RME.

## 4. Discussion

Loss-of-imprinting plays a crucial role in cancer development and progression, as it can lead to the abnormal activation or suppression of genes that are normally regulated based on their parental origin [5]. This aberrant expression is often linked to epigenetic mechanisms, particularly DNA methylation, which safeguards the monoallelic expression of imprinted genes. The removal of these methylation marks can result in the activation of the normally silent allele, ultimately leading to LOI in cancer [21]. This epigenetic alteration is often an early event in tumorigenesis and can contribute to tumor growth and metastasis. However, determining whether LOI is genuine can be challenging due to the complexity of imprinting mechanisms and the presence of biallelic expression patterns. Therefore, here, we leveraged the power of full-length single-cell RNA-sequencing to deepen our understanding of (loss-of) imprinting in breast cancer. The integration of SNP data from bulk RNA studies was, however, instrumental, as most SNPs featuring biallelic expression in the single-cell data could be attributed to transcript-level imprinting patterns, where paternally and maternally expressed transcripts share exons and SNPs. This also implies that full-length scRNA-seq is required, e.g., generated by the SMART technology, as solely relying on untranslated regions (typically 3′ UTRs) would be insufficient to provide a full overview of the transcriptional complexity.

By setting stringent criteria for biallelic expression, we have minimized the impact of sequencing errors and highlighted genuine LOI events. For instance, we observed clear LOI in *MEST* (also known as *PEG1*) in a patient’s HER2+ tumor cells, which was corroborated by bulk RNA-seq data from the same patient and the previous identification of tumor-specific LOI from TCGA bulk RNA-seq data. The LOI of *MEST* in cancer was already reported in breast cancer [22], adenocarcinoma [23], leiomyoma [24] and colorectal cancer [25]. Moreover, the observation that the LOI of *MEST* in mice leads to altered growth further supports its clinical relevance. However, it remained unclear whether this LOI was genuine. For example, in breast cancer, LOI was not associated with the anticipated overexpression of *MEST*, but with downregulation, leading to alternative hypotheses [4].

One such hypothesis relates to tissue-specific imprinting, where biallelic expression in normal cells infiltrating/present in the tumor mass leads to perceived LOI when *MEST* is downregulated in the tumor cells. Similarly, if imprinting-associated silencing is imperfect for *MEST*, resulting in (minor) residual biallelic expression, the downregulation of the expressed allele in cancer cells could lead to a relatively higher expression of the imperfectly silenced allele, contributing to perceived LOI [26]. Here, however, we demonstrated that *MEST* LOI indeed occurs in tumor cells, and that the level of expression is not lower than that for non-LOI samples/cells, showing that our observation was a genuine LOI event. An alternative explanation for the role of *MEST* LOI in cancer may hence be independent of the quantity of expression, and rather relate to alternative transcription, e.g., by promoter switching, as suggested by Pedersen et al. [27]. Also, for other genes featuring a combination of downregulation and LOI in cancer, e.g., for *ZDBF2* and *ZNF331*, also observed in cancer cells in this study, incorrect transcript usage should be further explored.

On the other hand, single-cell RNA-seq data can also identify incorrect LOI calls based on bulk data, as was observed for *LOC100294145*. Here, biallelic expression was clearly observed in normal immune cells rather than in cancer cells. This can be explained by the fact that *LOC100294145* is located in the human leukocyte antigen (HLA) gene cluster on chromosome 6, which features large-scale random monoallelic expression [4,26,28]. As this largely uncharacterized gene is reportedly non-coding, it cannot be a bona fide HLA gene, likely explaining why it was still included by Goovaerts et al. [4].However, based on scRNA-seq, it is clear that this gene is most likely regulated in a similar manner to its surrounding HLA genes.

Nevertheless, our integration of single-cell and bulk RNA-seq data still came with some limitations. For example, we completely relied on SNPs demonstrated to feature (loss of) imprinting in a previously reported bulk RNA-seq experiment, so we were unable to infer any reliable information about missing SNPs/genes (e.g., filtered out due to low allele frequency or coverage) or incorrect results from the latter study. Certain SNPs may not have been detected due to filtering criteria like the expression level or a low minor allele frequency. More importantly, the single-cell dataset was rather sparse in its number of samples, number of cells, and sequencing depth per cell. As a consequence, we may have missed infrequent LOI events (low prevalence), or LOI events for which the number of available cells or the sequencing depth of the involved genes were too low. For example, in several cases (genes and cancer types), biallelic expression was observed in the bulk RNA-seq but not in any single-cell RNA-seq from the same tumor specimen. To a certain extent, this could be mitigated through a pseudobulk approach, merging all cancer cells per patient for LOI detection, yet at the cost of the cell-level resolution required to evaluate LOI consistency over cells. Additionally, incorporating DNA sequence data would be particularly useful in differentiating between homozygous and heterozygous individuals for a given SNP, as only the latter can provide meaningful information about LOI.

The relatively low number of single cells analyzed in the original study can be explained by the more expensive and lower throughput SMART-seq protocol, at least when compared with, e.g., 10x scRNA-seq experiments focusing on the 3′UTR region. Alternative solutions, where single-cell platforms are coupled to single-molecule sequencing, or the introduction of novel, putatively less expensive full-length scRNA-seq technologies such as FLASH-seq [29], should further increase the sensitivity of scRNA-seq-based LOI detection. Ideally, this could lead to an integrated analysis of both bulk and single-cell RNA-seq data for LOI detection, conceptually similar to tools like Scissor [30]. Theoretically, bulk RNA-seq can even be completely removed from the equation, yet that would require single-cell data for hundreds of control and case samples, making the strategy proposed here far more cost efficient.

## 5. Conclusions

In conclusion, our proof-of-concept study demonstrates that integrating bulk RNA-seq and full-length scRNA-seq provides a solid framework that can be used to detect LOI in disease. In breast cancer, we demonstrated that LOI in *MEST* was genuine, even when featuring expression downregulation in bulk cancer data, supporting the idea that clinically relevant LOI is often associated with aberrant transcript usage rather than a higher expression of the imprinted gene per se.

## Figures and Tables

**Figure 1 biomolecules-14-01598-f001:**
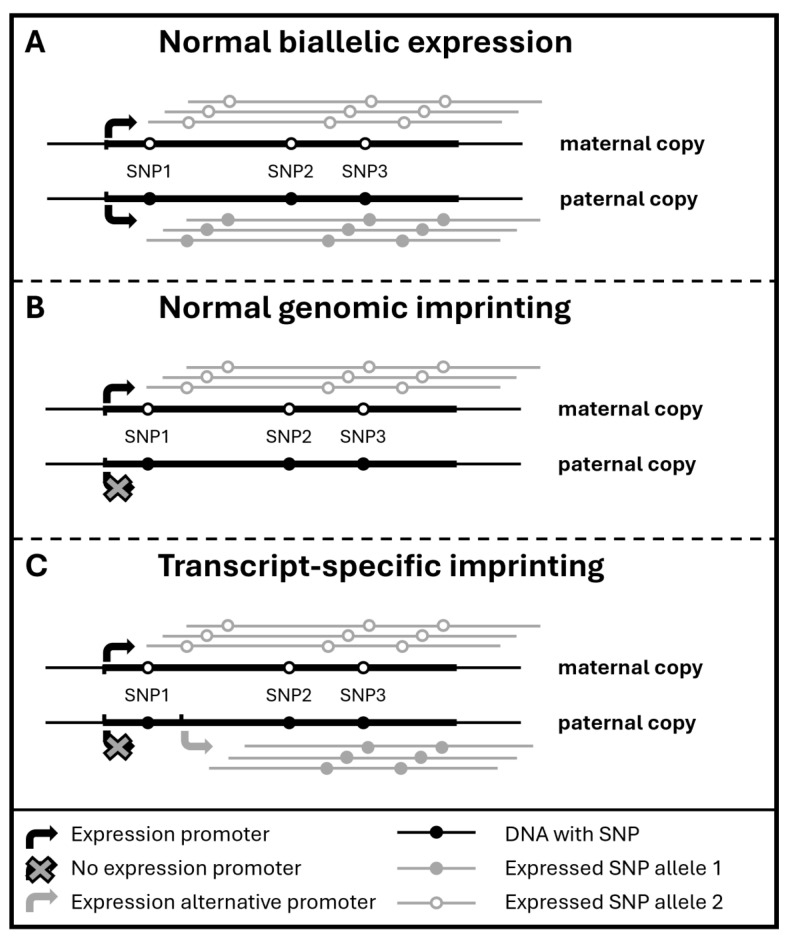
Genomic and transcript-specific imprinting. This figure illustrates three scenarios of gene expression, for an individual heterozygous for three SNPs. (**A**) Normal biallelic expression, where both copies of a gene are expressed, so that for all SNPs, both allelic variants are present in the transcripts. (**B**) Genomic imprinting, where only one parental copy (e.g., maternal) is expressed, leading to the expression of only the allelic variants inherited from that parent for all SNPs. (**C**) Transcript-specific imprinting, which occurs when some transcripts are exclusively paternally expressed and others exclusively maternally expressed, caused here by alternative promoter usage. This results in a mixed situation, with monoallelic expression for SNP1 but the biallelic expression of SNPs located in exons shared between maternal- and paternal-specific transcripts (SNP2 and SNP3). For simplicity, this depiction excludes splicing, though alternative splicing may also contribute to transcript-specific imprinting.

**Figure 2 biomolecules-14-01598-f002:**
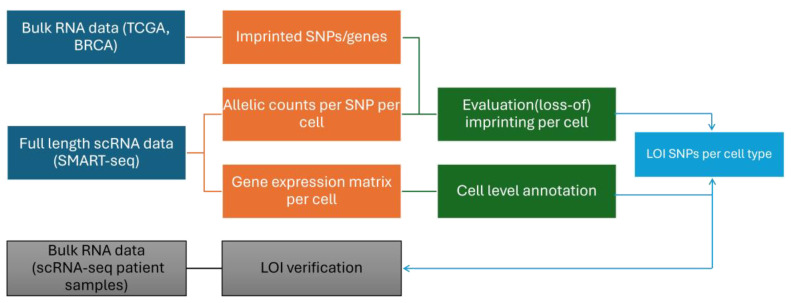
Overview. Strategy for the integration of bulk and single-cell RNA-seq data to identify genuine LOI in cancer. Note that verification using bulk RNA-seq data generated for the same patient tissues used for scRNA-seq is not essential for this strategy.

**Figure 3 biomolecules-14-01598-f003:**
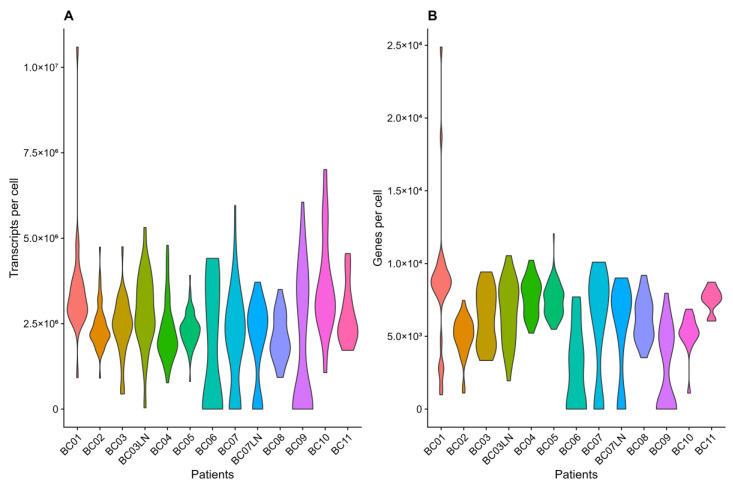
Comparative analysis of gene expression metrics in patient scRNA sequencing data. The left plot (**A**) displays the total transcript count per cell, while the right plot (**B**) visualizes the number of genes detected per cell. The width of each violin corresponds to the frequency of cells at a particular expression metric level, providing an intuitive understanding of the data’s underlying structure. Patient sample ID labels marked with an additional LN refer to metastatic samples obtained from the same patients.

**Figure 4 biomolecules-14-01598-f004:**
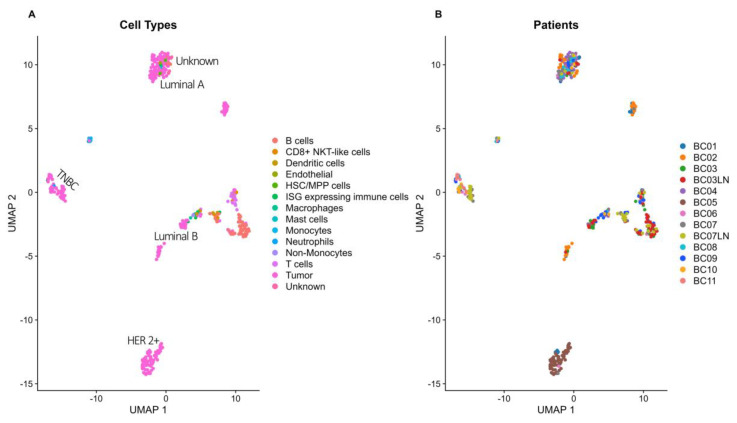
Comparative analysis of cell type distribution in tumor specimens from 11 patients. The left plot (**A**) visualizes the various cell types detected within the tumor specimens, with cells colored according to their inferred cell type. Unknown and tumor cell clusters are labeled as “unknown” or as the corresponding tumor subtype (luminal A, luminal B, HER2+ and TNBC). The right plot (**B**) displays the distribution of individual cells across the patients, highlighting cellular heterogeneity within the tumor samples. Yet when considering both panels simultaneously, patient cancer cells cluster per cancer subtype. Patient sample ID labels marked with an additional LN refer to metastatic samples obtained from the same patients.

**Figure 5 biomolecules-14-01598-f005:**
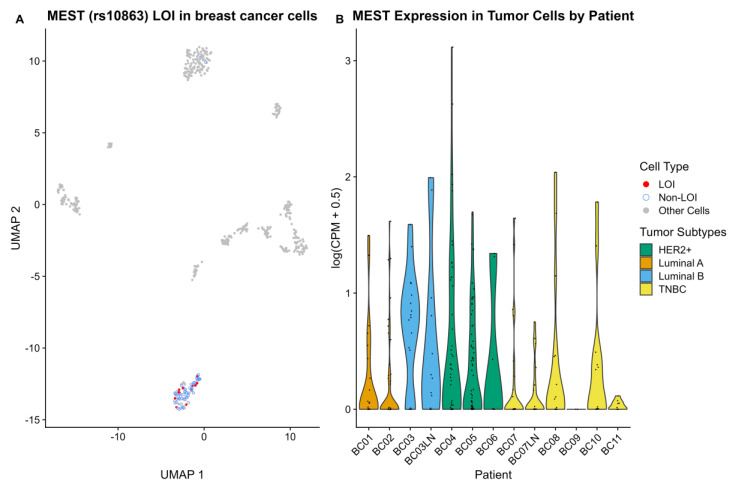
*MEST* expression across cells and patients. (**A**) Biallelic expression of rs10863 (*MEST* gene) in HER2+ tumor cells. Cells with clear LOI are indicated in red within the HER2+ tumor cluster for patient BC05, while non-LOI cells for that patient are shown in sky blue. Other cells are displayed in grey. Cells in the HER2+ tumor (BC05) cluster that are not labeled as LOI or non-LOI were either not identified as tumor cells or lacked sufficient SNP coverage. (**B**) *MEST* gene expression (log counts per million +0.5) across all patients, categorizing cells across different tumor subtypes indicating that LOI is not associated with *MEST* downregulation.

**Figure 6 biomolecules-14-01598-f006:**
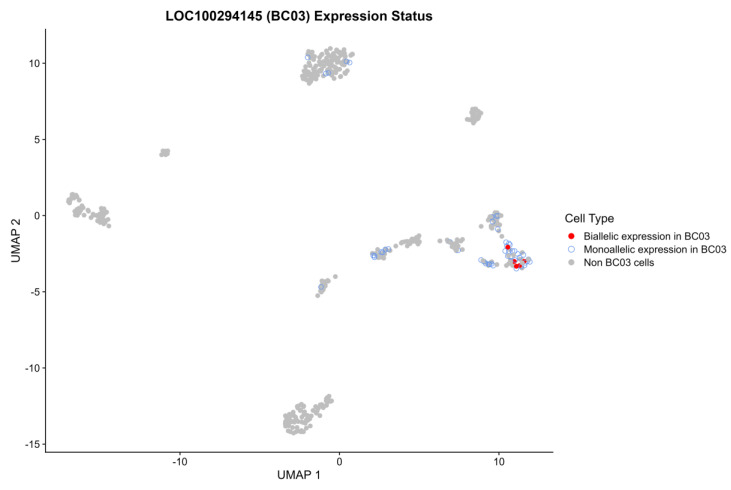
Biallelic expression of rs241402 and rs6912492 (*LOC100294145*) in immune B cells. Cells with biallelic expression (putatively RME) are indicated in red within the immune cells cluster for a metastatic luminal B patient. Other cells for that patient are shown in sky blue, and the rest of the cells are in grey.

**Table 1 biomolecules-14-01598-t001:** Annotation of cells.

Cell Types	Number of Cells
CD8+ NKT-like cells	2
HSC/MPP cells	1
ISG expressing immune cells	19
Dendritic cells	31
T cells	32
Neutrophils	2
Other immune cells	3
B cells	51
Tumor (luminal A, luminal B, HER2+, TNBC)	317 (108, 27, 109, 73)
Unknown	28
Total	486

**Table 2 biomolecules-14-01598-t002:** Imprinted SNPs detected by integrating scRNA and bulk RNA studies exhibiting significant biallelic expression (FDR < 0.05) in at least one cancer cell.

SNP Id	Gene	Location (Alleles)	LAF ^$^	Cell ID	TCGA Bulk LOI? *	Number of Single Cells	Estimate of LOI Fraction in Tumor Cells (%)	Detected in Same Patient Bulk (LAF)
rs10863	*MEST*	chr7:130505748 (A/G)	0.35	HER2+ tumor	Yes	9	75	Yes (0.21)
rs13073	*PEG10*	chr7:94667457 (T/C)	0.24	TNBC tumor	No	2	100	No
rs241402	*LOC100294145*	chr6:32901268 (T/C)	0.38	Pro-B cells	No	3	NA	Yes (0.26)
rs2839704	*H19*	chr11:1995429 (T/C)	0.25	HER2+ tumor	Yes	1	100	No
rs4673350	*ZDBF2*	chr2:206314148 (C/T)	0.2	HER2+ tumor	No	1	8.3	No
rs6912492	*LOC100294145*	chr6:32901932 (G/A)	0.27	Pro-B cells	No	2	NA	Yes (0.35)
rs73158510	*PTX3*	chr3:157443516 (G/A)	0.11	TNBC tumor	No	3	100	Yes (0.26)

$ After calculating the LAF for each locus, the average LAF was computed for cells exhibiting LOI. * Significant putative LOI (differential imprinting) in same cancer subtype according to Goovaerts et al. [4], using basal-like cancer as a proxy for TNBC tumors.

## Data Availability

The results published here are based upon data generated by European Nucleotide Archive (ENA) and are accessible upon application through the ENA Browser (ebi.ac.uk).

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
