# Peer review of "Characterization of Loss-of-Imprinting in Breast Cancer at the Cellular Level by Integrating Single-Cell Full-Length Transcriptome with Bulk RNA-Seq Data"

_biomolecules, 2024, doi:10.3390/biom14121598_

Round 1
Reviewer 1 Report
Comments and Suggestions for Authors
Amin et al are to be commended for demonstrating how valuable information about loss of imprinting can be extracted from existing single cell long read dataset. However, there are several issues in the manuscript which require clarification, together with some minor corrections.
1) While the results are convincing, the contribution of bulk RNA-Seq to the analysis should be more clearly explained. For example, it is stated that ‘two cancer cells from TNBC patient BC06 showed clear biallelic expression (lowest expressed allele detected 4 and 19 times, highest 19 and 53 times) for PEG10 270 (rs13073), which was not supported by the accompanying bulk sample’. A likely interpretation is that LOI occurs in so few tumour cells within the bulk sample that it is undetectable. This seems more plausible than concluding that it is not supported by the bulk sample.
2) The authors rightly highlight the key point that analysis of imprinting is complicated by its cell-type specificity. Single cell RNA sequencing is therefore the obvious approach to investigate. A simple strategy would be to compare expression (mono or biallelic) in tumour and normal cell types for all genes with available SNPs. For example, using ‘pseudobulk’ and pooling together all the sequence data from all the cells of each cell type from each individual? For imprinted genes one would expect to see biallelic expression in tumour cells and monoallelic in normal cells. Further clarification of why this is not adopted would be helpful.
3) Clarify the terminology, which is difficult to follow in many places. A definition of all the terms might avoid this confusion. For example:
I. Line 21: ‘biallelic expression arises from genomic imprinting at the transcript level; rather than loss-of-imprinting (LOI)’ To what does ‘genomic imprinting at the transcript level’ refer?
II. Clarify the statement (Line 34) that imprinting ‘is often transcript specific’. Does this refer to one isoform (splice variant) of a specific gene being imprinted but not other isoforms?
III. Line 68 states that single cell technologies targeting the UTR ‘do not provide the transcript-level resolution required for imprinting’. Clarify meaning – ie because sequences are not full length of transcript and therefore have reduced ability to detect SNPs? Or does this relate to discrimination of transcript isoforms?
IV. Line 167: ‘Similarly, SNPs biallelically expressed in both single-cell cancer and bulk normal RNA-seq data indicate the presence of transcript specific imprinting rather than LOI in cancer’. Clarify and explain rationale.
V. Line 216. ‘If imprinting is present, we would expect expression of only a single allele for all SNPs of that gene, with biallelic expression indicative for putative LOI. Note that the opposite does not automatically hold true, i.e. SNPs for which only a single allele is being detected may still feature (loss-of) imprinting when obscured by their homozygous status’ Clarify this paragraph by differentiating between interpretation of heterozygous and homozygous SNPs.
VI. Line 222: Many genes…‘featured SNPs with clear biallelic expression in multiple cancer cells, which is compatible with the basic definition of LOI. However, we hypothesized that this could be partially attributed to transcript specific imprinting, which is not captured by single-cell RNA seq’. Clarify what is meant by ‘transcript specific imprinting’ and why this is not captured by single-cell RNA-Seq.
Minor suggestions/corrections
3. It would be more appropriate to refer in abstract to the 486 cells that passed QC and were analysed rather than 549 initial cells.
4. Reduce use of semicolons in abstract
5. Line 234. In patient BC05 are there any non tumour cells and do they show imprinting, ie expression of both MEST alleles?
6. Line 68, 386: Misleading to say 10X only targets 3’UTR. Full length cDNA generated on the 10X platform can be used to generate libraries for sequencing on long read platforms (PacBio or Nanopore).
7. Comment further on the variability between samples. How could the distribution of samples BC06, BC07 and BC09 (Fig 2), with many cells truncated at the threshold of 200 genes mean for detection of LOI.
8. Line 281. Give abbreviation: random monoallelic expression (RME)
9. Figure 5 legend title: not Cell type (BC03 LOC100294145 expression status)
10. Suggest integrating the following sentences, which make the same point:
11. Line 332: ‘One such hypothesis was that there was residual biallelic expression in normal cells infiltrating/present in the tumor mass, which leads to perceived (increased) LOI in cancer when MEST is downregulated in the tumor cells. Similarly, if imprinting is imperfect, downregulation of the expressed allele in cancer cells would lead to relatively higher expression of the (imperfectly) silenced allele in cancer cells, and hence LOI.’
12. Ref 27 PARSE-seq does not seem to be appropriate for long read sequencing
13. Fig3 is coloured by cell type not tumour sub-type. Clarify that Tumour cells indicated by name of sub-type. Useful to label unknown. Do the patient cells match up completely with the sub-types (ie only cells from patient with specific subtype present in the cognate cluster?)
14. Discuss further the link between DNA methylation and LOI
15. It is mentioned, but I think important to acknowledge that DNA sequence would be helpful
Comments on the Quality of English Language
Quality of English language good with only a few minor errors
Reviewer 2 Report
Comments and Suggestions for Authors
Talal Amin et al describe a computational analysis focused on the effect of imprinting in cancer. Screening for imprinting is complicated. Here the authors integrate bulk and single-cell RNA-seq to investigate imprinting in breast cancer. The authors analyze a relatively small data set and focus the discussion on a single HER2+ tumor sample which exhibited loss of imprinting. I cannot recommend publication of the manuscript in its current format. I have the following major concerns:
1. Due to the relatively small sample size the generalizability of the observations is questionable. There are several potential technical confounders.
2. There are no statistics to assess the significance of the observations. Using the bulk RNA-seq to derive priors the authors need to apply statistical model to assess the single-cell RNA-seq data.
3. One potential approach to separate sequencing error from imprinting the authors could consider: Sequencing error will differ from allelic expression when looking at multiple SNPs covered by a single read. The two SNPs will occur together in the same read while sequencing error will not show any dependency. Can this approach be used to confirm the authors definition of putative sequencing errors?
4. The authors state: “a few neighboring SNPs (rs1050582, rs7658) demonstrated biallelic expression in this 243 sample’s tumor cells (Supplementary Table S5). Nevertheless, though the latter SNPs 244 were part of the same exon, it should be noted that their LOI status is less certain given 245 the absence of clear proof of imprinting in bulk data”. Is this a major concern? Why would the imprinting mechanism be restricted to part of an exon?
5. The authors should visualize the underlying counts and provide read coverage statistics in the tables. It is important to get a feeling for how robust the LAF estimates are regarding sequencing depth. Importantly, here only deduplicated reads should be counted. IGV screenshots would also increase the reader’s confidence in the reported results.
Comments on the Quality of English LanguageEnglish language is good.
Reviewer 3 Report
Comments and Suggestions for Authors
I have reviewed your manuscript "Characterization of Loss-of-Imprinting in Breast Cancer at the Cellular Level by Integrating Single-Cell Full-Length Transcriptome with Bulk RNA-Seq Data". Your study presents several innovative aspects, however I have identified some concerns that could affect the reproducibility and reliability of the findings.
My primary concerns are:
The integration between bulk and scRNA-seq data requires more robust validation. Tools like SCISSOR could leverage bulk RNA-seq signals to identify representative single cells, particularly for imprinted genes. This would strengthen the current analysis approach.
The LOI detection thresholds (≥4 counts, LAF > 5%) appear quite permissive. A validation using known imprinted genes could establish more robust cutoffs. The reliability of these findings would benefit from additional statistical confidence measures, please refer to these thresholds if they were derived from previous research.
The analysis of non-tumor cells is not shown. Examining the imprinting patterns in matched normal tissue and stromal cells would provide crucial context. Given the importance of the tumor microenvironment, immune cell imprinting analysis would be particularly informative.
Your dataset could benefit from CNV integration. This would not only validate tumor/normal classifications but could reveal relationships between copy number changes and LOI events.
Regarding reproducibility, while your methods section (Lines: 154-155) mentions custom scripts, having these publicly available rather than "upon reasonable request" would be valuable. This would enable validation of LOI detection thresholds, bulk/single-cell data integration methods, Normalization strategies and Cell-type classification approach
Addressing these points would enhance the impact of your valuable work in the field.
Round 2
Reviewer 1 Report
Comments and Suggestions for Authors
I am glad that comments were helpful and agree that the manuscript is now much improved. Most queries have been addressed.
A clear understanding of transcript-specific imprinting is essential to appreciate the results of the study. It might be helpful to include a schematic diagram indicating transcript-level imprinting. Although this issue has been addressed in the response to comments, there is still confusion in places requiring clarification:
1. Line 222: ‘Similarly, SNPs biallelically expressed in both single-cell cancer and bulk normal RNA-seq data indicate the presence of transcript-specific imprinting rather than LOI in cancer.’
This is confusing because SNPs biallelically expressed in both single-cell cancer and bulk normal RNA-seq data indicate genes which are not imprinted. Are you meaning that another possible explanation is that they are imprinted at the level of individual transcripts? In this case it should state that this ‘MAY indicate the presence of transcript-specific imprinting’ (for which you would need full length RNA-Seq data to determine)?
2. Line 320: ‘As our short-read (Illumina) based scRNA-seq data does not allow for transcript-level analyses’
But the scRNA-seq is full-length SMART-Seq libraries. Is the reason unable to do transcript level analysis of scRNA-Seq data not more to do with limited number of reads per cell? Also clarify that bulk RNA-Seq that captures reads from the full-length of the transcriptome (or long reads) is required for identifying transcript-specific imprinting. Some bulk RNA-Seq protocols focus on the 3′ end of transcripts and therefore suffer the same inability to discriminate transcript isoforms as standard scRNA-Seq (10X).
The first paragraph on page 7 now explains clearly how apparent biallelic expression of SNPs can in fact be due to transcript-specific imprinting if located in a shared exon. The remainder of this paragraph is difficult to follow – please consider rewording. For example, does ‘SNP level resolution imprinting data’ refer to multiple SNPs that enable discrimination of different transcripts and therefore biallelic expression/transcript-specific imprinting?
Previous comments:
Comment 7. Fewer genes per cell in a sample reflects fewer reads per cell and therefore fewer reads per gene. I intended a comment on the fact that it is therefore more difficult to detect LOI in those samples with fewer reads.
Comment 13. Fig3A still not quite correct - The cells are coloured by cell type and the clusters are labelled by tumour subtype
Considering the modified text, which now includes this point later, suggest deleting this statement. Line 387: In other cases, LOI could not be readily verified in the matching bulk RNA sample for that patient or was considered less relevant for the tumor subtype under study.
